



# Technical note: Transit times of reactive tracers under time-variable hydrologic conditions

Raphaël Miazza[1] and Paolo Benettin[1]

[1]Institute of Earth Surface Dynamics, Faculty of Geoscience and the Environment, Université de Lausanne, Lausanne, Switzerland

**Correspondence:** Raphaël Miazza (raphael.miazza@unil.ch)

**Abstract.** Water transit time distributions (TTDs) have been widely used in hydrology to characterize catchment behavior. TTDs are also widely used to predict tracer transport, but the actual transit times of tracers, which may differ from those of water because of different physical processes and tracer input patterns, remain largely unexplored. Here, we address the TTDs of tracers transported by water and subjected to linear processes of sorption, degradation and interaction with evapotranspira-

tion. We focus on the special case of randomly sampled systems (which are mathematically similar to well-mixed systems), for which analytical solutions can be derived. Through the analytical solutions and their numerical implementation under time-variable flow conditions, we explore how reactive transport parameters impact tracer TTDs. Results show that sorption delays tracers as much as a larger water storage does. Evapotranspiration can both increase tracer transit times (in the case of evapoconcentration) or decrease them (in the case of net evaporation extraction), while degradation can be seen as an additional

output flux that always shortens tracer transit times. Combinations of randomly-sampled systems are widely used as transport models and we show how tracer TTDs may differ from water TTDs in the building blocks of such models. Distinguishing the TTDs of tracer from those of water is important for an improved understanding of water quality dynamics and the circulation of solutes at the catchment scale.

## 1   Introduction

The time water spends moving through the landscape fundamentally influences the transport and release of solutes to streams (Li et al., 2021). Therefore, understanding how catchments store, mix, and release water through the lens of transit times is useful for interpreting water quality dynamics (Hrachowitz et al., 2016), such as predicting contaminant transport and nutrient export to streams. Water transit time (TT), usually defined as the time interval between the entrance of a water parcel into a catchment as precipitation and its exit as streamflow or evapotranspiration (Botter et al., 2010), is often described through

distributions (TTD) that reflect the large heterogeneity in flowpaths and velocities that characterize the subsurface environment. TTDs are usually described mathematically through various classes of models, including: compartment models (Metzler et al., 2018; Botter, 2012; Zenger and Niemi, 2009), Lagrangian approaches based on stochastic advective-reactive processes or mass response functions (Cvetkovic and Dagan, 1994; Cvetkovic et al., 2012; Botter et al., 2005; Rinaldo et al., 1989), and (ground)water age equations based on the dispersion model (Ginn et al., 2009; Ginn, 1999; Cornaton and Perrochet, 2006).



While usually developed independently, these classes of models have clear connections among them (see Benettin et al., 2013a; Leray et al., 2016).

  In this study we focus on TTDs from compartment models, because they are a useful descriptor of catchment-scale hydrological processes that can be directly linked to tracer measurements in streamflow. In this context, TTDs and stream tracer concentration $C_Q$ are coupled through a convolution integral:

$$C_Q(t) = \int\limits_0^{+\infty} C_S(T,t)\, p_Q(T,t)\, \mathrm{d}T, \tag{1}$$

where $t$ is time, $T$ is transit time, $C_Q(t)$ is the stream concentration series, $C_S(T,t)$ is the concentration of a water parcel entered at time $t - T$ after traveling (and possibly reacting) for a time interval $T$, and $p_Q(T,t)$ is the time-variable TTD. The term "tracer" is used here in a general sense to refer to solutes and isotopes, whether conservative or reactive. The convolution integral (Eq. 1) has a dual use: it allows us to infer TTDs from tracer concentration series (inverse problem, see McGuire and

McDonnell, 2006), or it allows us to use the TTDs as a transport model and predict tracer concentration in streamflow (direct problem, see Nguyen et al., 2021). The parcel's tracer content during transport to the outlet ($C_S(T,t)$) is typically modeled by applying reactions on top of the initial parcel's concentration. Examples of reactions include linear decay (Małoszewski and Zuber, 1982), evapoconcentration (Bertuzzo et al., 2013), first-order kinetics (Druhan and Benettin, 2023). Many research papers used water TTDs as a transport model along with some reactive treatment of the tracer to simulate the transport of

reactive tracers like nitrate (Nguyen et al., 2022; Yu et al., 2023), dissolved organic carbon (Grandi et al., 2025), pesticides (Bertuzzo et al., 2013; Lutz et al., 2017), silicon (Benettin et al., 2015a) and tritium (Wang et al., 2023).

  The TTD developments highlighted above have been mainly carried out by hydrologists, and so they can be seen as 'water-centric' in the sense that the ultimate goal has usually been that of inferring TTDs from tracer series and use them as catchment descriptors. While a great expansion in the theoretical characterization of *water* TTDs occurred in the last 15 years (Benettin

et al., 2022), the transit times of *tracers* carried out by water, which are generally expected to be different from those of water, have not been addressed explicitly. Even when tracer TTD equations have been developed (Harman and Xu Fei, 2024), they have been used to compute tracer content, not to investigate tracer transit times. There is thus an opportunity to learn how different processes influence tracer transit times in hydrological systems. Tracer transit time can be defined as the time interval between the entrance of tracer mass into a system and its exit through any output (streamflow, root water uptake but

also reaction). Physical processes like sorption, degradation and the interaction with vegetation may cause tracers to spend longer/shorter time in a catchment than the water carrying them. Consequently, water transit time may not be a good predictor of the time tracers have to interact with the environment.

  Here, we address the time-variable TTDs of reactive tracers transported by water under the simplifying condition of a well-mixed system–or more precisely, a system that is 'randomly sampled' by the outflows. Although a single randomly sampled

system may not be a realistic representation of any real-world landscape, combinations of randomly sampled systems arranged in series and in parallel form the basis of many lumped and distributed transport models (e.g. Remondi et al., 2018; Kuppel et al., 2018; Hrachowitz et al., 2015; Rodriguez et al., 2018; van Huijgevoort et al., 2016). Therefore, this study aims to





uncover the tracer TTDs dynamics at the core of such models. Our specific goals are to 1) Derive new analytical solutions addressing the TTDs of reactive tracers and 2) Use these expressions to investigate how the processes of sorption, degradation

and evapoconcentration influence tracer transit times and their partitioning to different pathways. Distinguishing the TTDs of tracer from those of water is important for an improved understanding of stream solute sources and water quality dynamics at the catchment scale.

## 2   Starting points

### 2.1   Water age equations

The general time-variable theory of water TTDs in catchments mainly originates from the works of Botter et al. (2011), van der Velde et al. (2012) and Harman (2015), and was later expanded (Rigon et al., 2016) and summarized in a review work (Benettin et al., 2022).

The starting point is considering some simple hydrologic system, in which a 'control volume', representing e.g. a catchment, a hillslope or a soil plot, is characterized by a water storage ($S$) that evolves over time $t$ in response to an input flux $J(t)$, which

may represent precipitation, and two output fluxes: streamflow $Q(t)$ and evapotranspiration $ET(t)$. Mass conservation results in the hydrologic balance equation:

$$\frac{dS(t)}{dt} = J(t) - Q(t) - ET(t) \tag{2}$$

with some initial condition $S(0) = S_0$.

For any parcel of water entering through the input flux $J$, we can define the parcel's age $T$ as the time elapsed since its

entrance into the system. When the principle of water mass conservation is extended to incorporate the age dimension, the 'water age balance' can be expressed as:

$$\frac{d[S(t)\overleftarrow{p_S}(T,t)]}{dt} = J(t)\delta(T) - Q(t)\overleftarrow{p_Q}(T,t) - ET(t)\overleftarrow{p_{ET}}(T,t) \tag{3}$$

with initial condition $\overleftarrow{p_S}(T, t=0) = \overleftarrow{p_{S_0}}(T)$ and null boundary condition. Here, $\overleftarrow{p_S}$ represents the age distribution of water in storage, while $\overleftarrow{p_Q}$ and $\overleftarrow{p_{ET}}$ represent the age distributions of water leaving the system via discharge and evapotranspiration,

respectively. The input water flux $J(t)$ is associated with a Dirac delta distribution $\delta(T)$, reflecting the fact that by definition incoming water has an age of zero (Botter et al., 2011).

Equation (3) is usually solved numerically after introducing a 'StorAge Selection' (SAS) function, which is a mathematical expression linking the outflow and storage age distributions (Botter et al., 2011; van der Velde et al., 2012; Harman, 2015). A special case is the one where all outflows are made of a uniform (or random) sample of the stored water parcels. This condition,

which is mathematically equivalent to a 'well-mixed' system, leads to a simple analytical solution, which is useful to explore how the hydrologic fluxes and storage of a system influence the transit times through it. Under a uniform-sampling condition, all age distributions are identical: $\overleftarrow{p_S}(T,t) = \overleftarrow{p_Q}(T,t) = \overleftarrow{p_{ET}}(T,t)$. By substituting this condition into Eq. 3 and removing



the input term (to be treated as a Dirichlet boundary condition $\overleftarrow{p_S}(T=0,t) = J(t)/S(t)$), Eq. 3 can be developed into the first-order partial differential equation:

$$\frac{\partial \overleftarrow{p_S}(T,t)]}{\partial t} + \frac{\partial \overleftarrow{p_S}(T,t)]}{\partial T} = -\frac{Q(t) + ET(t) + \frac{dS(t)}{dt}}{S(t)} \overleftarrow{p_S}(T,t) \tag{4}$$

which can be solved analytically (see Appendix A and Polyanin et al., 2001) for $\overleftarrow{p_S}$ and thus for $\overleftarrow{p_Q}$ and $\overleftarrow{p_{ET}}$ (Botter et al., 2011; Botter, 2012). The solution reads:

$$\overleftarrow{p_Q}(T,t) = \overleftarrow{p_S}(T,t) = \frac{J(t-T)}{S(t)} \exp\left(-\int_{t-T}^{t} \frac{Q(\tau) + ET(\tau)}{S(\tau)} d\tau\right) \tag{5}$$

As the system is not at hydrologic steady state, the mean transit time changes over time. The water long-term mean transit time ($w\overline{MTT}$) can be well approximated by the mean of $\overleftarrow{p_Q}$ at steady state, which is simply the ratio between the system's long-term storage and fluxes:

$$w\overline{MTT} = \frac{\overline{S}}{\overline{Q + ET}} \tag{6}$$

which is valid for any system, not just randomly sampled ones.

While Eq. 5 describes the "backward" TTD, i.e. the distribution of ages of water parcels leaving the system at time $t$ via a given outflow, transit time distributions can also be defined in a "forward-in-time" sense, by defining transit times with respect to a group of water parcels entering a system at injection time $t_i$ and leaving the system at subsequent time $t = t_i + T$ in a particular outflow. This transit time distribution is defined as the forward transit time distribution and indicated as $\overrightarrow{p_Q}$ (Harman, 2015). The forward transit time distribution can be computed by developing and solving a "forward" water age balance equation (Benettin et al., 2015b). Alternatively, one can obtain it from the continuity relationship that links forward and backward TTDs (Niemi, 1977; Botter et al., 2011), which for a system with one input ($J$) and multiple outputs (e.g. $Q$ and $ET$) reads:

$$J(t-T)\theta_Q(t-T)\overrightarrow{p_Q}(T,t-T) = Q(t)\overleftarrow{p_Q}(T,t) \tag{7}$$

where the parameter $\theta_Q$ is a partitioning coefficient, representing the fraction of precipitation that entered at time $t - T$ and whose fate was to leave via streamflow. The forward TTD, though less popular than the backward TTD, is useful for interpreting breakthrough curves. To ease the notation, we use the entrance time $t_i$ of a water parcel and express the current time as $t = t_i + T$. By considering a water parcel entering the system at time $t_i$, its normalized breakthrough curve (wNBTC) to streamflow is the fraction of the parcel's initial mass which is found in streamflow at subsequent lag times $T$. The wNBTC is obtained by inserting Eq. 5 in Eq. 7:

$$wNBTC(T,t_i) = \theta(t_i)\overrightarrow{p_Q}(T,t_i) = \frac{Q(t_i+T)}{S(t_i+T)} \exp\left(-\int_{t_i}^{t_i+T} \frac{Q(\tau) + ET(\tau)}{S(\tau)} d\tau\right) \tag{8}$$

Note that, because not all the mass ends up in the streamflow output, the wNBTC is not a probability density function as it does not integrate up to one. Rather, it will integrate to the total fraction of initial mass that ends up in streamflow, which is $\theta_Q(t_i)$.

Together, Eqs. 5–8 provide a complete description of water TTDs from precipitation to streamflow and form the basis for the development of the new equations addressing tracer TTDs.





## 2.2 Tracer mass balance

For the same randomly sampled system described in Section 2.1, we can now consider tracer mass fluxes and relate them to
the mass in storage through a mass conservation equation. The input mass entering the control volume is termed $\dot{m}_I(t)$, and,
importantly, it does not need to enter via a water input like precipitation. The tracer mass can exit the storage via streamflow
($\dot{m}_Q(t)$) and evapotranspiration ($\dot{m}_{ET}(t)$). In addition to transport processes, biogeochemical reactions and decay may alter
the tracer mass within storage. The transformation rate due to such reactions is expressed as $\dot{m}_R(t)$. The changes in the stored
tracer mass $M_S$ are thus expresses as:

$$\frac{dM_S(t)}{dt} = \dot{m}_I(t) - \dot{m}_Q(t) - \dot{m}_{ET}(t) + \dot{m}_R(t) \tag{9}$$

starting from an initial condition $M_S(0) = M_{S_0}$. Figure 1a illustrates the system.

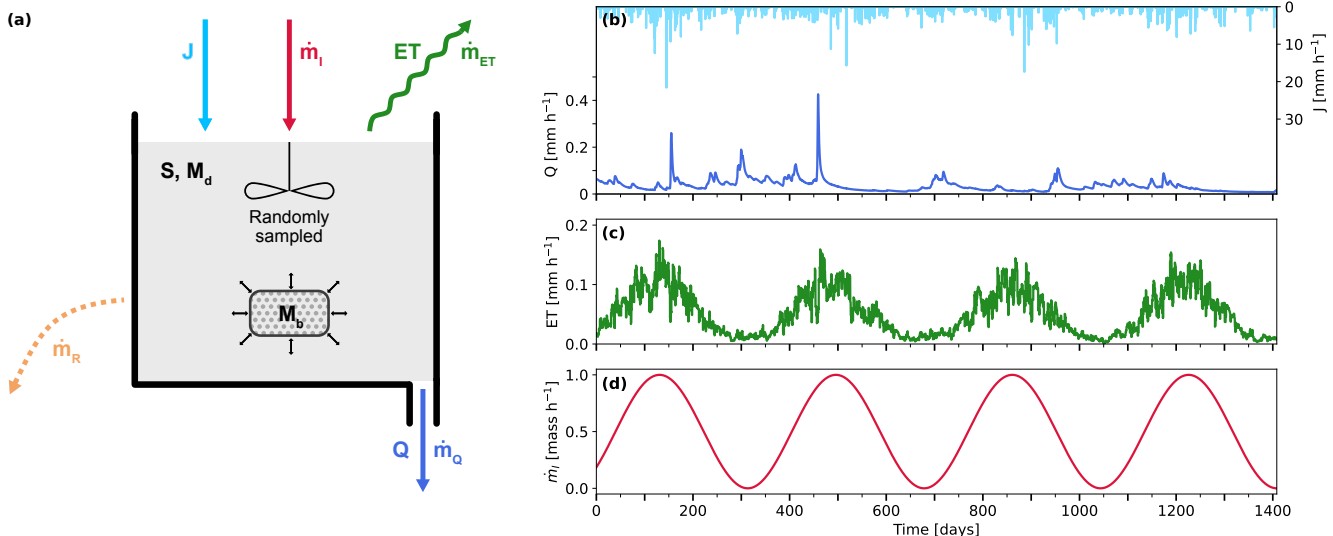

**Figure 1.** Schematic representation of the hydrological system under consideration **(a)** with example water fluxes **(b, c)** and tracer fluxes
**(d)**. The approach requires series of rainfall ($J$), discharge ($Q$), evapotranspiration ($ET$) and input tracer mass ($\dot{m}_I$), and computes all other
mass fluxes based on the random sampling assumption and specified linear transformations.

We assume that mass transport occurs solely via water movement. For simplicity, we also assume that mass inputs not
associated with precipitation (e.g., dry deposition or dry fertilizer applications) are instantaneously dissolved into the existing
water storage. Alternative dissolution models (e.g. through chemical kinetics) can be used at no loss of generality. Because
water fluxes are assumed to randomly sample the water in storage, their tracer concentrations reflect the tracer concentration
in storage ($C_S(t)$), but we consider three processes that make tracer transport different from a pure passive transport: sorption,
degradation and evapoconcentration.





We account for sorption by considering that tracer mass in storage can be present in two states: a dissolved mobile state ($M_d$), and a sorbed immobile state ($M_b$), such that $M_S(t) = M_d(t) + M_b(t)$. The dissolved mass in storage is expressed through a classic approach accounting for linear, reversible and instantaneous sorption: $M_d(t) = M_S(t)/R$, where $R$ is usually termed a "retardation" factor in the Advection-Dispersion Equation literature (Genuchten, 1982) and can be computed as a function of soil bulk density ($\rho$), volumetric water content ($w$) and the distribution ratio between sorbed and dissolved concentration ($K_d$) as $R = 1 + K_d \rho / w$ (see Bertuzzo et al., 2013). When considering sorption, the tracer concentration in storage corresponds to the ratio between the dissolved tracer mass and the water storage volume:

$$C_S(t) = \frac{M_d(t)}{S(t)} = \frac{1}{R}\frac{M_S(t)}{S(t)} \tag{10}$$

While we focus on sorption here, the retardation factor framework can in principle be used for processes like anion exclusion (Gerritse and Adeney, 1992) and take values $R < 1$.

We consider a linear mass degradation that applies to both the dissolved and sorbed phase:

$$\dot{m}_R(t) = -k \cdot M_d(t) - k \cdot M_b(t) = -k \cdot M_S(t) \tag{11}$$

where $k$ is a decay constant that can be conveniently expressed in terms of the tracer's half life (DT50) as $k = \ln(2)/\mathrm{DT}50$. Linear decay formulations have been widely used for the transport of radioactive isotopes (Małoszewski and Zuber, 1982; Morgenstern et al., 2010; Rodriguez et al., 2021) and for degradable compounds (e.g., Queloz et al., 2015).

We consider that the tracer concentration in the mass lost through evapotranspiration, $C_{ET}$, is proportional to the tracer concentration in storage through an 'evapoconcentration' constant $\alpha \geq 0$:

$$C_{ET}(t) = \alpha \cdot C_S(t) \tag{12}$$

For $\alpha = 0$, tracer mass is completely excluded by evapotranspiration, thus increasing the storage concentration. For any value $0 < \alpha < 1$, tracer mass is partly left behind by evapotranspiration, which also increases the storage concentration. $\alpha = 1$ corresponds to the case of a tracer passive to $ET$, while values $\alpha > 1$ indicate that the tracer is preferentially extracted (e.g. a nutrient) and leads to a decrease in storage concentration. Formulations similar to Eq. 12 have frequently been used in the literature (e.g., Harman and Xu Fei, 2024) to model the evapoconcentration of water stable isotopes (Benettin et al., 2017) and solutes like agricultural chloride (Benettin et al., 2013b; Harman, 2015), pesticides (Bertuzzo et al., 2013) and other reactive solutes (Druhan and Bouchez, 2024). Following Eq. 12, the mass output in the $ET$ flux is $\dot{m}_{ET}(t) = \alpha \cdot ET(t) \cdot C_S(t)$.

Other types of linear reactions could be added to the mass balance. For example, the dissolution of solutes originating from mineral weathering like silicon (Druhan and Bouchez, 2024) could be modeled through first-order kinetics and appear as an additional input term.

## 3 New tracer transit time solutions

We define tracer age as the time elapsed since the incoming tracer mass dissolved into the system's storage. The dissolution time coincides with the time of entry when the tracer is already in dissolved form or dissolves instantaneously, but it may differ



when dissolution is delayed. Based on this definition, the tracer age distribution $\overleftarrow{\rho}(T,t)$ is the distribution of tracer mass with
respect to age $T$ at time $t$. This concept applies to the tracer mass stored within the system (either in dissolved or sorbed form)
and to that leaving the system.

The random sampling assumption introduced in Section 2 implies that all water age distributions are identical to the storage
age distribution, and thus, the streamflow concentration equals the storage concentration. Under this assumption, it can be
shown that tracer mass is also uniformly sampled, leading to equal mass age distributions across storage and fluxes: $\overleftarrow{\rho_Q}(T,t) =$
$\overleftarrow{\rho_{ET}}(T,t) = \overleftarrow{\rho_R}(T,t) = \overleftarrow{\rho_S}(T,t)$. This relationship holds even in the presence of reactions that may alter tracer concentration
during transport through the system.

We introduce a tracer age balance, which is analogous to the concept of water age balance, describing the evolution of tracer
mass within the system as a function of both time and age. Under the random sampling assumption, the tracer age balance
corresponding to the mass balance in Eq. 9 is given by:

$$\frac{d[M_S(t)\overleftarrow{\rho_S}(T,t)]}{dt} = \dot{m}_I(t)\delta(T) - \dot{m}_Q(t)\overleftarrow{\rho_S}(T,t) - \dot{m}_{ET}(t)\overleftarrow{\rho_S}(T,t) + \dot{m}_R(t)\overleftarrow{\rho_S}(T,t) \tag{13}$$

with initial condition $\overleftarrow{\rho}_S(T,t=0) = \overleftarrow{\rho_{S_0}}(T)$ and null boundary condition. The input tracer mass flux $\dot{m}_I$ is again associated
with a Dirac delta function $\delta(T)$, as by definition the new tracer mass dissolving into the system has age zero. The mass age
balance described by Eq. 13 may appear similar to the one introduced by Harman and Xu Fei (2024), but it has an important
structural difference in the way that age is tracked. While the mass age balance by Harman and Xu Fei (2024) addresses the
mass (potentially of different ages) associated with a water parcel of a given age, our balance addresses directly the mass of a
given age (regardless of the age of the water it is associated with). In many cases, these two approaches coincide in practice.
However, they are notably different in case of dissolution processes: new dissolving mass is associated with (water of) multiple
ages in the approach of Harman and Xu Fei (2024), while it is only associated with an age of zero in our tracer-oriented balance.

Incorporating Eqs. 10–12 into Eq. 13, and moving the input mass to the boundary condition $\overleftarrow{\rho}_S(T=0,t) = \dot{m}_I(t)/M_S(t)$,
Eq. 13 can be developed into the first-order partial differential equation:

$$\frac{\partial \overleftarrow{\rho_S}(T,t)}{\partial t} + \frac{\partial \overleftarrow{\rho_S}(T,t)}{\partial T} = -\left( \frac{Q(t) + \alpha \cdot ET(t)}{R \cdot S(t)} + k + \frac{\frac{dM_S}{dt}}{M_S(t)} \right) \overleftarrow{\rho_S}(T,t) \tag{14}$$

which can be solved analytically (see Appendix A and Polyanin et al., 2001). The general solution is:

$$\overleftarrow{\rho_Q}(T,t) = \overleftarrow{\rho_S}(T,t) = \frac{\dot{m}_I(t-T)}{M_s(t)} \exp\left[ -\int_{t-T}^{t} \left( \frac{Q(\tau) + \alpha \cdot ET(\tau)}{R \cdot S(\tau)} + k \right) d\tau \right] \tag{15}$$

Both the water TTD (Eq. 5) and the tracer TTD (Eq. 15) have an equivalent formula where the input fluxes rather than output
fluxes appear inside the integral (see Appendix A). However, such formula does not allow one to see the model parameters
explicitly and so it is not reported. Because the hydrologic and mass fluxes may vary over time, the mean tracer transit time
will change over time as well, but the long-term mean tracer transit time $m\overline{MTT}$ can be approximated through the mean of
$\overleftarrow{\rho_Q}$ at steady state:

$$m\overline{MTT} = \frac{R \cdot \overline{S}}{\overline{Q} + \alpha \cdot \overline{ET} + k \cdot R \cdot \overline{S}} \tag{16}$$



The mean transit time does not have much meaning in practical applications because it's very difficult to estimate reliably (Kirchner, 2016a; Benettin et al., 2022), but it's useful here for interpreting the analytical expressions.

To derive the formula for the mass breakthrough curve in streamflow we proceed as we did with the water breakthrough curve, by invoking the Niemi relationship (Eq. 7) applied to the mass age distributions. We term $\theta_{\dot{m}_Q}$ the partitioning coefficient that quantifies the fraction of mass entering the system at time $t_i$ that will ultimately exit the system via streamflow. The function

$\overrightarrow{\rho_Q}(T, t_i)$ denotes the "forward" TTD for tracer mass entering at time $t_i$ and leaving through streamflow (which is formally equal for all output fluxes in a randomly sampled system). The result for the mass normalized breakthrough curve is:

$$mNBTC(T, t_i) = \theta_{\dot{m}_Q}(t_i)\overrightarrow{\rho_Q}(T, t_i) = \frac{Q(t_i + T)}{R \cdot S(t_i + T)} \exp\left[-\int\limits_{t_i}^{t_i + T}\left(\frac{Q(\tau) + \alpha \cdot ET(\tau)}{R \cdot S(\tau)} + k\right) d\tau\right] \qquad (17)$$

Eq. 15 and Eq. 17 provide a complete description of tracer transit times through the simplified hydrological system described here. They are the tracer counterparts to the water transit time distributions Eq. 5 and Eq. 8 and as such they are the starting

point to explore how tracer TTDs are different from water TTDs.

## 4 Insights from the Analytical Expressions

The water and mass age equations (Eq. 4 and Eq. 14) are first-order, linear, partial differential equations whose solutions (Eq. 5 and Eq. 15) are exponential distributions with time-variable exponents that depends primarily on the hydrologic fluxes and storage.

In the case of an ideal passive tracer–that is, one that does not undergo sorption, evapoconcentration or decay ($R = 1$, $\alpha = 1$, $k = 0$)–the breakthrough curves for water and tracer mass (Eq. 8 and Eq. 17) coincide. This is expected as tracer mass in our system moves transported by water. However, even if each mass application moves exactly like water, the age distributions (i.e., backward TTDs) of water and tracer are not the same because the input timeseries (or more precisely, the timeseries of the ratio between input and storage) are different. This is clearly visible when comparing Eq. 5 and Eq. 15, which only differ in

the term in front of the exponential. As the age distribution in streamflow represents the contribution of past inputs to present streamflow, different input patterns will generate different distributions. Concrete examples with seasonal tracer input patterns are given in Section 5.1.

Another insight that emerges is that one does not need tracer data to compute the breakthrough curves $mNBTC$ (Eq. 17) nor the long-term mean transit time $mMTT$ (Eq. 16), as they are entirely determined by the hydrologic variables ($Q$, $ET$

and $S$) and the transport parameters ($R$, $\alpha$ and $k$). The effect of such parameters can thus be investigated theoretically. The retardation factor $R \geq 1$, which is used to reduce the total tracer mass storage to the soluble mass storage, always appears as a multiplier of the water storage $S$ and can be virtually interpreted as a storage magnifier. Thus, the effect of sorption is effectively equivalent to the effect of a larger storage: it will delay the tracer and result in longer tracer transit times. The evapoconcentration parameter $\alpha$ can be seen as a valve that controls the mass directed to $ET$. When $\alpha < 1$, less mass will go

to $ET$, resulting in tracer accumulation in storage and thus longer transit times. Conversely, when $\alpha > 1$ (net tracer extraction)





the larger tracer output rate will result in a depleted tracer storage, with faster turnover and shorter transit times. Degradation can be seen just as an additional tracer output flux controlled by the kinetic rate constant $k$ [1/T]. Because of this additional exit pathways, which results again in a depletion of the storage mass and faster turnover, transit times through the system are reduced. Therefore, the outflowing mass will generally be younger when degradation is faster.

While the theoretical analysis of tracer TTDs proves already insightful, further explorations under realistic, time-variable conditions requires numerical implementation.

## 5   Insights from a Numerical Implementation in Time-Variable Flow Conditions

The system described in Sections 2–3 was implemented numerically in Python and solved using a fixed hourly time step. Any series of hydrologic fluxes and input tracer data can be used to run the model. Here, we use as a case study the hydrologic data

from Duchemin et al. (2025) because it is a ready to use and realistic hydrologic series. An excerpt of the data is shown in Fig. 1b, c. The rainfall timeseries correspond to hourly intensity rates recorded at the Basel (Switzerland) meteorological station run by the Swiss Federal Office of Meteorology and Climatology (MeteoSwiss). Over the selected time period, the average annual rainfall amount is 806 mm. The output water fluxes (Q and ET) were simulated by Duchemin et al. (2025) through a three-compartment hydrological model, which is based on the two-box framework presented by Kirchner (2016b, 2019). The

yearly average ET and Q equal 449 mm and 357 mm, respectively. The initial storage $S_0$ was set equal to the average annual rainfall amount (806 mm) to induce a mean transit time of water equal to 1 year.

The water balance (Eq. 2) was solved using the hydrologic time series by Duchemin et al. (2025). The tracer mass balance (Eq. 9) was solved using a fourth-order Runge-Kutta method. The input mass flux ($\dot{m}_I$) was defined as a sine wave with an annual frequency (Fig. 1d), representing an arbitrary tracer subject to seasonally variable inputs (e.g., fertilizers, pesticides,

or water stable isotopes). The sine wave was set to peak in summer and reach its minimum in January. While input mass flux timeseries were provided to the model, the output mass fluxes ($\dot{m}_Q$, $\dot{m}_{ET}$ and $\dot{m}_R$) were computed dynamically during the numerical integration. This was achieved by evaluating the intermediate stages of the state variables $M_S(t)$ and $S(t)$ within the Runge-Kutta 4 scheme, ensuring that the overall mass balance was preserved. All fluxes were assumed constant within each time step, while state variables were defined at the beginning of each time step. To attenuate the influence of initial conditions

on the simulated output mass fluxes, $M_0$ was set equal to its steady-state value based on the average hydrological fluxes and mass input.

The numerical implementations of the water and tracer age distribution (Eqs. 5 and 15), as well as the normalized water and tracer mass breakthrough curve (Eqs. 8 and 17) were performed after computing the time series of water storage $S(t)$. These equations require a second layer of discretization, as they depend not only on time $t$ but also on transit time $T$, which must

also be discretized. The discretization of transit time is aligned with the time stepping of the model. For the age distributions, the first transit time bin, $T_{j=0}$, corresponds to water or tracer mass that entered and exited the system within the same time step $t$. Similarly, for mNBTCs, $T_{j=0}$ corresponds to tracer mass that entered at time $t_i$ and exited within the same time step. Subsequent transit time steps represent progressively longer transit times, defined in increments equal to the model time





step (i.e., one hour). All data and code used or generated in this case study are available in a publicly accessible repository
(https://github.com/rmiazza/tracer-transit-times).

## 5.1 Passive tracers

As shown theoretically in Sect. 4, if the input timeseries of water and passive tracers differ, these differences will be reflected
in the age distributions. We illustrate these discrepancies in Fig. 2, by considering a seasonally variable input timeseries of
passive tracer mass ($R = 1$, $\alpha = 1$, $k = 0$) to the system (Fig. 1d), with larger mass inputs in the summer and lower mass
inputs in the winter. This seasonal variability may be representative of seasonal patterns in water stable isotope content or
agricultural products application. While the overall time-averaged water and tracer TTDs appear similar (Fig. 2a), as they
reflect approximately steady-state conditions, the individual (time-varying) TTDs vary considerably, as shown by the thin lines
in Fig. 2a. Moreover, the input tracer seasonal cycle influences the TTD of tracer mass over different times of the year. During
the winter months (Fig. 2b), when tracer inputs are low, most of the tracer mass leaving the system originates from inputs that
occurred during previous high-input periods (i.e. in summer). This is reflected in the time-averaged tracer TTD which shows
most of its weight around a transit time of approximately 200 days. In contrast, during summer (Fig. 2c), a large fraction of
mass in or leaving the system will be relatively young, due to the high input rates of (necessarily young) tracer at that time.
Because the water input fluxes in this study exhibit less seasonality than the tracer inputs, such patterns are not observed in the
water TTDs. As a result, even for passive tracers, the transit times of water and tracer leaving the system differ under these
conditions.

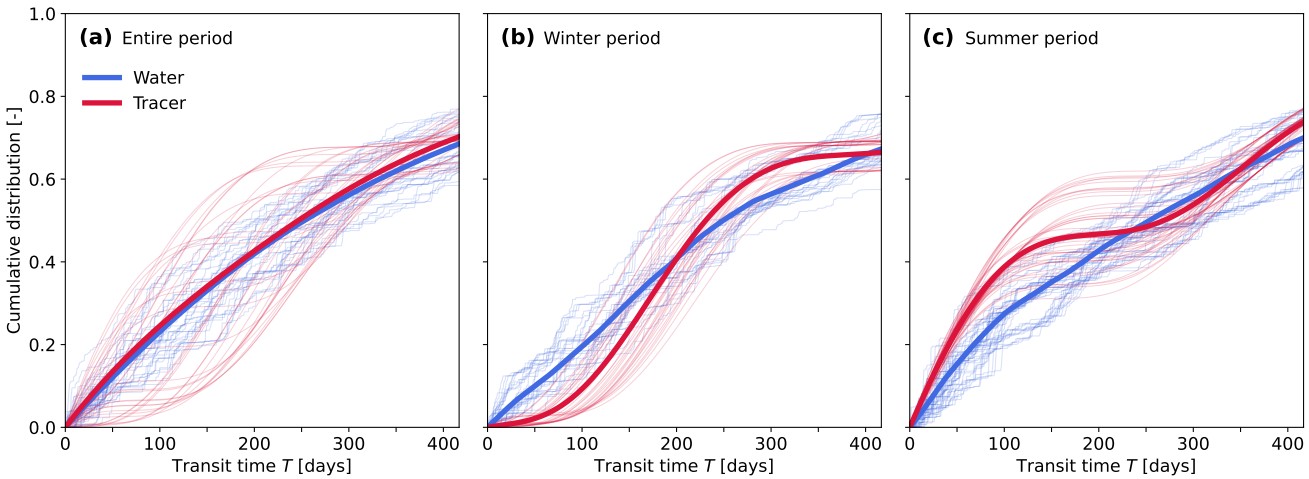

**Figure 2.** Age distributions (TTD) for water and passive tracer mass leaving the system during **(a)** the entire time period, **(b)** winter and **(c)**
summer. The time-averaged TTD is shown as a thick line, while thin lines represent individual TTDs at selected time steps.





## 5.2 Non-passive tracers

Based on Eq. 15, it is clear that tracer-specific characteristics influence the shape of tracer age distributions. To illustrate this effect, we compute the overall time-averaged age distributions by varying one parameter at a time, allowing us to isolate and visualize the influence of each parameter separately (Fig. 3). As expected, increasing the retardation factor leads to longer

transit times, as reflected by the flattening of the TTDs for higher retardation values in Fig. 3a, compared to the TTD of water. The increased sorption effectively slows the movement of tracer mass through the system. Tracer mass fluxes leaving through evapotranspiration have the potential to slow down ($\alpha < 1$) or accelerate ($\alpha > 1$) the tracer movement through the system compared to water (Fig. 3b). As explained in Sect. 4, this behavior arises from how the evapoconcentration factor influences the total mass stored in the system, ultimately affecting the tracer's turnover rates. Finally, while ET can both accelerate or

decelerate the tracer's transport through the system, degradation consistently accelerates tracer transport relative to water, potentially by large factors depending on the decay rate associated with the tracer (Fig. 3c). Similar to the effect of high mass rates extracted by ET ($\alpha > 1$), this behavior is explained by the reduction in total tracer mass stored in the system under high decay rates, which increases the overall turnover and shortens transit times.

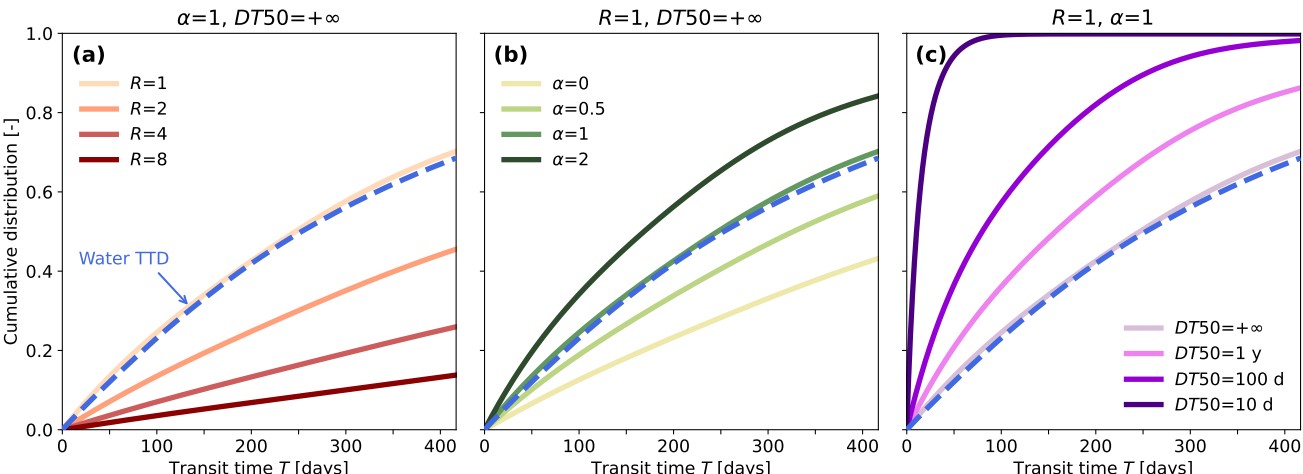

**Figure 3.** Time-averaged age distributions for water and tracer mass leaving the system, illustrating the effect of different tracer characteristics. **(a)** Increasing tracer age with increasing retardation factor. **(b)** Tracer age decreases with increasing evapoconcentration factor ($\alpha > 1$) and increases with decreasing evapoconcentration factor ($\alpha < 1$). **(c)** Decreasing tracer age with decreasing $DT50$ value. In each panel, the two non-varying parameters are held constant at their passive tracer values.

## 5.3 Tracer breakthrough curves

While age distributions reflect the distribution of transit times of tracer mass leaving the system at a given time–strongly shaped by the history of input time series–the tracer breakthrough curves in streamflow (mNBTC) describe the distribution of transit





times for a tracer mass parcel that entered the system at a specific time. In addition, mNBTCs reflect the fraction of that tracer mass that ultimately reached streamflow at subsequent time steps. They are therefore influenced both by the overall velocity of tracer mass through the system and by how that mass is partitioned among the different outflow fluxes. To assess how different 295  tracer characteristics influence the mNBTCs, we computed the mNBTC for every individual tracer mass input across the study period. Following the approach in Sect. 5.2, we varied each of the three tracer parameters separately, keeping the other two fixed at their passive-tracer values. Results for a single, arbitrary input time step are presented in Fig. 4, where panels 4a–c show the mNBTCs, and panels 4d–f display the corresponding cumulative mNBTCs for the same tracer mass injection, thereby illustrating the progressive recovery of the input mass in streamflow over time.

Individual, time-varying mNBTCs are typically irregular (Fig. 4a–c), reflecting the dynamic nature of the hydrological system considered here. As previously shown for water TTDs by Botter et al. (2010), discharge plays a key role in shaping the tracer breakthrough curves, as it is a multiplier of the exponent in Eq. 17 and regulates the exported mass. A discharge timeseries is reported for comparison in Fig. 4a. The transport parameters affect tracer transport in intuitive ways. An increase in sorption (higher retardation factor $R$) visibly slows the release of tracer mass to the stream (Fig. 4a), as shown by the flattening of 305  the mNBTC. This effect is solely due to a reduction in tracer velocity through the system as the mass partitioning among the different outputs is unchanged (because the other parameters are held at their passive-tracer settings). All curves shown in Fig. 4d will reach the same plateau of 0.41, corresponding to the partitioning term $\theta_{\dot{m}_Q}$ of the considered mass application, although this is not apparent in Fig. 4 due to the cutoff at 400 days. In contrast, increasing the fraction of mass that is extracted by ET ($\alpha > 1$, Fig. 4b,e) accelerates turnover through the system but reduces the tracer recovery in streamflow, leading to a 310  breakthrough curve that is always lower than that of the passive tracer. The effect of changing $\alpha$ only becomes relevant when ET fluxes are important. This is visible in Fig. 4b,e where summer starts around day 100. Finally, higher degradation rates (lower $DT50$) decrease tracer recovery in streamflow and shorten transit times due to faster mass turnover (Fig. 4c,f).

Tracer application (or tracer 'labeling') experiments are routinely used to investigate transport processes in soils. In some cases, contrasting breakthrough curves have been reported for different tracers (e.g. Benettin et al., 2019; Groh et al., 2018), 315  likely due to differences in sorption, degradation, or interactions with root water uptake. The analytical breakthrough curves developed here, especially when arranged in parallel or in series to move beyond a single randomly sampled unit, are useful for the quantitative interpretation of empirical data from tracer experiments, particularly when multiple tracers are used.

## 5.4 Mass partitioning to multiple outputs

While tracer characteristics influence their transit times through the system, they also affect the ultimate fate of the tracer mass 320  entering the system. As described in Sect. 5.3, these characteristics shape the tracer breakthrough curves in streamflow, but more generally, they influence how mass is ultimately partitioned among the different output fluxes. A convenient way to explore this partitioning is to compute the partitioning coefficients. Because the system is dynamic, these partitioning coefficients will differ for each individual mass input. Accordingly, we compute the three partitioning coefficients ($\theta_{\dot{m}_Q}$, $\theta_{\dot{m}_{ET}}$, and $\theta_{\dot{m}_R}$) corresponding to the three output mass fluxes, for every tracer mass input. The partitioning coefficients were computed by







**Figure 4.** Tracer normalized breakthrough curves (mNBTC) illustrating the effect of different tracer characteristics on the release of tracer mass to streamflow. Panels a–c: mNBTC corresponding to an individual input timestep for varying values of the retardation factors **(a)**, evapoconcentration factor **(b)** and $DT50$ **(c)**. Panels d–e: cumulative mNBTC corresponding to the same individual input timestep, illustrating the recovery of tracer mass in streamflow over time. The discharge time series following this input is shown as an inset in panel **(a)**. In each panel, the two non-varying parameters are held constant at their passive tracer values.

integrating the breakthrough curves for discharge, evapotranspiration and degradation fluxes over all transit times $T$ (see formulas in Appendix B).

     Figure 5 presents box plots of the three partitioning coefficients computed over the entire time window, for four cases with different combinations of tracer-characteristic parameters. The spread of the partitioning coefficients in the box plots is primarily driven by variations in hydrological fluxes and storage, and is therefore largely influenced by the specific time series

used in this study. In the first case (Fig. 5a), the parameter set is chosen such that the partitioning coefficients are approximately balanced, with the tracer behaving as a passive tracer in all respects except for its decay. In the second case (Fig. 5b), the





retardation factor is increased, leading to a significant shift in tracer mass partitioning towards decay. This outcome reflects the fact that the combination of strong sorption and degradation is particularly effective at removing mass from the system via decay, at the expense of the two other output fluxes. This effect is also evident in the steady-state analytical expressions for

partitioning coefficients (Appendix B), where the product of the decay rate $k$ and the retardation factor $R$ directly influences the partitioning toward reactive loss. The third case (Fig. 5c) represents a scenario with prominent evapoconcentration (i.e., $\alpha << 1$). As a consequence, the mass partitioned toward evapotranspiration drops sharply, while partitioning toward both streamflow and decay increases. In the final case (Fig. 5d), the $DT50$ value is doubled, effectively halving the decay rate. As expected, this leads to a substantial decrease in the partitioning toward decayed mass, accompanied by increases in the fractions

reaching streamflow and evapotranspiration. More generally, Figs. 5c–d illustrate that similar tracer recovery in streamflow can result from very different internal processes–high evapoconcentration in one case, and low degradation in the other.

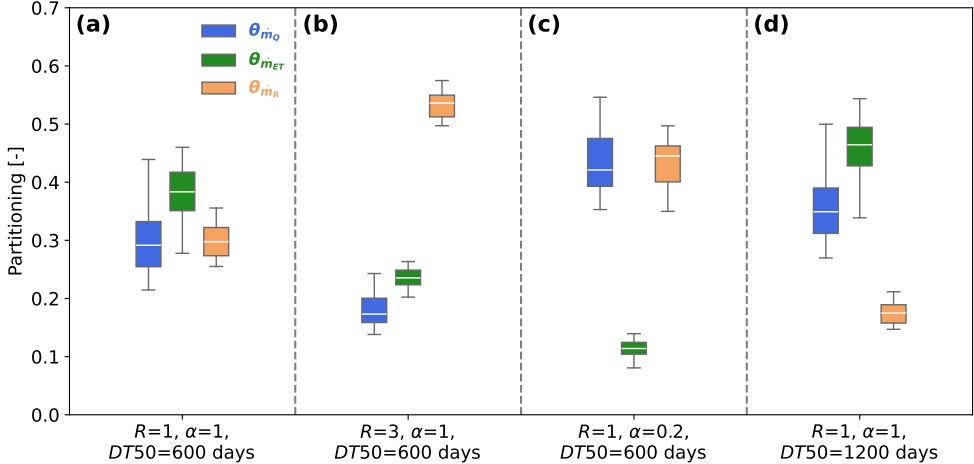

**Figure 5.** Box-plots of the partitioning coefficients for each tracer mass input, showing the fraction directed to streamflow (blue), evapotranspiration (green), and degraded (orange). The boxes represent the interquartile range, with the median shown as a white line. Whiskers extend to the minimum and maximum values within 1.5 times the interquartile range from the lower and upper quartiles, respectively. Results are shown for four parameter configurations: **(a)** reference case, **(b)** increased retardation factor, **(c)** reduced evapoconcentration factor, and **(d)** increased $DT50$.

## 6  Conclusions

Our work leads to the following conclusions:

- Processes like linear tracer sorption, degradation and evapoconcentration can be easily included into a tracer age balance
equation. Randomly sampled systems support an analytical solution for the tracer age distribution that is similar to that of the water age distributions.





– Such analytical solution builds on the framework initiated by Botter et al. (2010) and extends it to the case of reactive tracers.

– Even a perfectly passive tracer that behaves like water will generally have an age distribution that differs from that of water because of the difference between water and tracer input timeseries.

– Transport models based on transit time distributions have rarely accounted for mass sorption processes. Here, we demonstrate how sorption can affect tracer transit times showing that its effect is analogous to that of increased storage.

– Tracers are on average younger than the water carrying them when they are affected by degradation or there is a net extraction by ET fluxes. They are instead older than water when there is sorption and evapoconcentration.

– Those same physical processes not only affect the velocity at which tracers move through a system, but also their recovery through the different outputs.




**Appendix A: Analytical solution of the age balance equations in the case of random sampling**

The solutions to the water age balance and the tracer age balance equations are structurally similar, differing only by certain coefficients. To unify the derivations, we introduce a general notation and provide a common formulation for both cases. Let
$y$ denote the transit time distribution to be solved, expressed as a function of both transit time $T$ and clock time $t$. With this notation, Eqs. 4 and 14 can be rewritten as:

$$\frac{\partial y(T,t)}{\partial t} + \frac{\partial y(T,t)}{\partial T} = f(t)y(T,t) \tag{A1}$$

where $f(t) = -\frac{Q(t)+ET(t)}{S(t)} - \frac{dS(t)/dt}{S(t)}$ in the case of Eq. 4, and $f(t) = -\frac{Q(t)+\alpha \cdot ET(t)}{R \cdot S(t)} - k - \frac{dM_S(t)/dt}{M_S(t)}$ in the case of Eq. 14. We can express both cases more generally as $f(t) = -A(t) - \frac{dB(t)/dt}{B(t)}$. In the case of the water age balance, $A(t)$ corresponds
to the ratio of total outflows to water storage, and $B(t)$ is the water storage $S(t)$. For the tracer age balance, $A(t)$ additionally includes the effects of sorption, evapoconcentration and degradation, while $B(t)$ corresponds to the tracer mass in storage $M_S(t)$. Following Polyanin et al. (2001), the general solution to this first-order partial differential equation reads:

$$y(T,t) = \Phi(t-T)\exp\left[\int_0^t f(\tau)d\tau + C\right] \tag{A2}$$

with $\Phi(t-T)$ being an arbitrary function determined by the boundary conditions and $C$ the integration constant. After applying
the boundary condition $y(T=0,t) = g(t)$ specific to each case ($g(t) = \frac{J(t)}{S(t)}$ and $g(t) = \frac{\dot{m}_I(t)}{M_S(t)}$, respectively), Eq. A2 becomes:

$$y(T,t) = g(t-T)\exp\left[\int_{t-T}^t f(\tau)d\tau\right] \tag{A3}$$

The exponential term on the r.h.s. of Eq. A3 can be further developed:

$$\exp\left[\int_{t-T}^t f(\tau)d\tau\right] = \exp\left[\int_{t-T}^t \left(-\frac{\frac{dB(\tau)}{d\tau}}{B(\tau)} - A(\tau)\right)d\tau\right]$$

$$= \exp\left[-\int_{t-T}^t \frac{\frac{dB(\tau)}{d\tau}}{B(\tau)}d\tau\right]\exp\left[-\int_{t-T}^t A(\tau)d\tau\right] \tag{A4}$$

$$= \frac{B(t-T)}{B(t)}\exp\left[-\int_{t-T}^t A(\tau)d\tau\right]$$

Inserting Eq. A4 into Eq. A3 results in the fully derived analytical solutions of the water and tracer age balances in randomly sampled systems (Eqs. 5 and 15).

Alternatively, Eqs. 5 and 15 can be equivalently reformulated using inflows instead of outflows in the exponential term. This equivalent formulation arises from rewriting the general term $f(t)$ using the continuity equations (Eqs. 2 and 9), leading to





$f(t) = g(t) = \frac{J(t)}{S(t)}$ for the water age balance and $f(t) = g(t) = \frac{\dot{m}_I(t)}{M_S(t)}$ for the tracer age balance. Substituting these expressions
into the general solution (Eq. A2) and applying the boundary conditions leads to:

$$\overleftarrow{p_Q}(T,t) = \frac{J(t-T)}{S(t-T)} \exp\left(-\int_{t-T}^{t} \frac{J(\tau)}{S(\tau)} d\tau\right) \tag{A5}$$

$$\overleftarrow{\rho_Q}(T,t) = \frac{\dot{m}_I(t-T)}{M_S(t-T)} \exp\left(-\int_{t-T}^{t} \frac{\dot{m}_I(\tau)}{M_S(\tau)} d\tau\right) \tag{A6}$$

While these solutions are more compact, the equivalent solutions Eq. 5 and especially Eq. 15 are more convenient for exploring
the effect of hydrologic and transport parameters on TTDs.

**Appendix B: Derivation of normalized mass breakthrough curves and partitioning coefficients for tracer mass extracted through evapotranspiration or degraded**

The analytical formula for normalized breakthrough curves in streamflow was derived from the Niemi relationship (Niemi, 1977; Botter et al., 2011), which expresses continuity for tracer mass in input and in streamflow over both time and age.
Similarly to water (Eq. 7), the Niemi relationship for tracer mass in streamflow can be expressed as:

$$\dot{m}_I(t-T)\theta_{\dot{m}_Q}(t-T)\overrightarrow{\rho_Q}(T,t-T) = \dot{m}_Q(t)\overleftarrow{\rho_Q}(T,t) \tag{B1}$$

Since the system is randomly sampled, forward TTDs are the same for all output fluxes (the same applies to the backward TTDs, as shown in Sect. 3). Thus, $\overrightarrow{\rho_Q}(T,t-T) = \overrightarrow{\rho_{ET}}(T,t-T) = \overrightarrow{\rho_R}(T,t-T)$, where $\overrightarrow{\rho_{ET}}(T,t-T)$ and $\overrightarrow{\rho_R}(T,t-T)$ represent the forward TTDs of mass entering at time $t-T$ and leaving through evapotranspiration or being degraded, respectively. More
generally, the statement of mass conservation over time and age can be expanded to all output mass fluxes. For mass leaving through ET, the Niemi relationship reads as:

$$\dot{m}_I(t-T)\theta_{\dot{m}_{ET}}(t-T)\overrightarrow{\rho_{ET}}(T,t-T) = \dot{m}_{ET}(t)\overleftarrow{\rho_S}(T,t) \tag{B2}$$

and for degraded mass, it reads as:

$$\dot{m}_I(t-T)\theta_{\dot{m}_R}(t-T)\overrightarrow{\rho_R}(T,t-T) = \dot{m}_R(t)\overleftarrow{\rho_S}(T,t) \tag{B3}$$

Since mNBTC are defined as the product between the partitioning coefficient and the forward TTD, these curves can be obtained by rearranging Eqs. B2 and B3. Rewriting these equations in terms of $t = t_i + T$, and using Eq. 15, the mNTBC can be expressed as:

$$mNBTC_{\dot{m}_{ET}}(T,t_i) = \theta_{\dot{m}_{ET}}(t_i)\overrightarrow{\rho_{ET}}(T,t_i) = \frac{\alpha \cdot ET(t_i+T)}{R \cdot S(t_i+T)} \exp\left[-\int_{t_i}^{t_i+T}\left(\frac{Q(\tau)+\alpha \cdot ET(\tau)}{R \cdot S(\tau)} + k\right)d\tau\right] \tag{B4}$$



$$mNBTC_{\dot{m}_R}(T,t_i) = \theta_{\dot{m}_R}(t_i)\overrightarrow{\rho_R}(T,t_i) = k \cdot \exp\left[-\int\limits_{t_i}^{t_i+T}\left(\frac{Q(\tau)+\alpha \cdot ET(\tau)}{R \cdot S(\tau)}+k\right)d\tau\right] \tag{B5}$$

where $mNBTC_{\dot{m}_{ET}}$ and $mNBTC_{\dot{m}_R}$ represent the normalized breakthrough curve of mass leaving through ET and degraded tracer mass, respectively. From Eqs. 17, B4 and B5, it follows directly that integrating the mNTBC over the entire age domain yields the partitioning coefficient associated with a tracer mass input at time $t_i$ (since the integral of the forward TTD equals unity).

Additionally, long-term mean partitioning coefficients can be computed from the average hydrological and mass input fluxes. Let $\overline{Q}$, $\overline{ET}$, and $\overline{S}$ denote the average streamflow, evapotranspiration and water storage, respectively. By replacing the time-varying fluxes with their steady-state (i.e., time-averaged) counterparts in Eqs. 17, B4, and B5, and integrating over transit time for $T \rightarrow +\infty$, the following expressions for the steady-state partitioning coefficients are obtained:

$$\overline{\theta}_{\dot{m}_Q} = \frac{\overline{Q}}{\overline{Q}+\alpha \cdot \overline{ET}+k \cdot R \cdot \overline{S}} \tag{B6}$$


$$\overline{\theta}_{\dot{m}_{ET}} = \frac{\alpha \cdot \overline{ET}}{\overline{Q}+\alpha \cdot \overline{ET}+k \cdot R \cdot \overline{S}} \tag{B7}$$

$$\overline{\theta}_{\dot{m}_R} = \frac{k \cdot R \cdot \overline{S}}{\overline{Q}+\alpha \cdot \overline{ET}+k \cdot R \cdot \overline{S}} \tag{B8}$$

*Code and data availability.* During peer review, the code used in this paper is available on the github repository https://github.com/rmiazza/
tracer-transit-times. Upon revision, it will be published on the FAIR-compliant Zenodo repository. The data for the case study used in the numerical computations are available from Duchemin et al. (2025).

*Author contributions.* **Raphaël Miazza:** conceptualization, software, formal analysis, visualization, writing – original draft. **Paolo Benettin:** conceptualization, supervision, formal analysis, writing – original draft.

*Competing interests.* The authors declare no competing interests

*Acknowledgements.* The authors thank the Faculty of Geoscience and the Environment of the University of Lausanne for financial support.



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
