# Peer review of "Technical note: Transit times of reactive tracers under time-variable hydrologic conditions"

_EGUsphere, 2025_

## Referee Comment (RC2)

**Review of: "Technical note: Transit times of reactive tracers under time-variable hydrologic conditions" by Raphael Miazza and Paolo Benettin**

This technical note describes how the transit time approach developed for tracking water ages through catchments can be extended to yield transit times of reactive tracers. I find this paper a well written and valuable addition to the theory development around transit time distributions of water and solutes. This study applies the theory of transittime distributions for water to solutes and combines it with assumptions from advective-diffusion solute transport modelling (Retardation: constant equilibrium between dissolved and sorbed concentration, linear decay, fractionation). As such it is a novel contribution. Below I list a couple of suggestions that in my opinion could improve the manuscript:

Overall:

The paper has an unconventional structure: introduction, starting point, New solutions, insights from these new solutions, insights from Numerical implementation of these new solutions, conclusions. Consider a more conventional setup with: Introduction (1), Theoretical development (2, containing the old 2,3&4), Example application(3) methods/setup (3.1), results (3.2), discussion (3.3) and conclusion (6). I can see why you chose your setup: you don't want to put a strong focus on the model implementation and it is not the goal in itself, but now section 5 reads cramped and its goal is unclear.

In the theory development and discussion I miss what your results mean for not fully mixed (i.e preferential flow systems) systems. It looks to me that without the full mixing-assumptions you can almost fully copy the same reasoning and your derivations will also hold for preferential flow systems. Could you not do the theory development for any system and then do your example (section 5) for the fully mixed one? That would make the structure a bit easier to read as well? You can easily write the SoluteMass-Age balance eq:

$$\frac{\partial M(t-s,t)}{\partial t} = M_I(t)\delta(t-s=0) - \frac{M(t-s,t)}{RS(t-s,t)}Q(t-s,t) - \frac{\alpha M(t-s,t)}{RS(t-s,t)}E(t-s,t)$$
$$- k\,M(t-s,t)$$

Which solves to:

$$p_{Mq}(t-s,t) = \frac{M(t-s,t)}{M(t)} = \frac{M_I(s)}{M(t)}\exp\left(-\int_s^t \left[\frac{Q(t-s,\tau)+\alpha E(t-s,\tau)}{RS(t-s,\tau)}+k\right]d\tau\right)$$

In fact, we can only derive water transit time distributions through following tracers (=solutes) through the system. Thus basically, all previous publications that first derive Pq

$$C_Q(t) = \int_0^\infty C_J(t - T) p_Q(T, t) dT$$

and subsequently use                                                 must have already implicitly applied a Solute transit time distributions, just with R=1, alfa =1 and k = 0 (stable isotopes). I think you can probably show both approaches are exactly the same thing. But you are much deeper into this and can reason better why or why not. I can be convinced that this all can be tackled through extra discussion on mixed systems.

I would suggest to write the conclusions more engaging less bullet-wise

Line 223

"The evapoconcentration parameter α can be seen as a valve that controls the mass directed to ET. When α < 1, less mass will go to ET, resulting in tracer accumulation in storage and thus longer transit times. Conversely, when α > 1 (net tracer extraction) the larger tracer output rate will result in a depleted tracer storage, with faster turnover and shorter transit times." Although entirely correct, I find the valve quite abstract as plats don't have valve but can fractionate at there roots. I would suggest α < 1 for solutes that plants try to keep out of there system (high chloride, toxic solutes), α > 1 for solutes that plants need to for their functioning (Nutrients, trace elements)

Figure 5: Caption it would help add the partition coefficient variables to the caption (I was scanning the paper for θmR, but couldn't find it until close reading the caption)

Conclusion 2 is not clear and to me not really a conclusion: "Such analytical solution builds on the framework initiated by Botter et al. (2010) and extends it to the case of reactive tracers"

**Unrelated but I had too much fun adding a dissolution term. Maybe for next paper**

Just for fun adding a source term that could represent weathering. Not sure if it works :

$$\frac{\partial M(t-s,t)}{\partial t} = M_I(t)\delta(t-s=0) - \frac{M(t-s,t)}{RS(t-s,t)}Q(t-s,t) - \frac{\alpha M(t-s,t)}{RS(t-s,t)}E(t-s,t)$$

$$- k\,M(t-s,t) + j\left(\frac{M(t-s,t)}{RS(t-s,t)}\text{-C0}\right)$$

$$M(t-s,t) = M_I(s)\exp\left(-\int_{\tau=s}^{t}\left[\frac{Q(t-s,\tau)+\alpha E(t-s,\tau)-j}{RS(t-s,\tau)}+k\right]d\tau\right)$$

$$- jC_0\int_{\xi=s}^{t}\exp\left(-\int_{\tau=\xi}^{t}\left[\frac{Q(t-s,\tau)+\alpha E(t-s,\tau)-j}{RS(t-s,\tau)}+k\right]d\tau\right)d\xi$$

---

## Author Comment (AC1)

**Response to Referee #1**

Please find below our responses to Referee #1, with the original comments in black and our responses in blue.

The paper addresses the non-stationary theory of transit time distribution under simplified assumptions, derives analytical solutions for a simplified case, explains them, and presents some conceptual applications. Overall, I believe the paper has merit. The concepts developed, while not entirely novel, have not yet broken through to the broader hydrological community's consciousness. However, the applications require clarification of several aspects to be truly useful. I find value in the paper, but for the sake of reproducibility, I recommend that the authors more clearly specify the conditions under which the simulations were conducted. I recommend minor revisions.

We are grateful to Referee #1 for their time to review our manuscript and for the positive evaluation of our work. We hope that the revisions made in response to Referee #1's specific comments will improve the clarity of the numerical implementation of our theoretical developments.

Detailed Comments:

Page 8 - Equation (17): Is the so-called "mass-normalized breakthrough curve" not simply the discharge at time T+t_i generated by the input at time t_i? Additionally, how can one determine the partition coefficient at any finite time? More practically, how can this be approximated given that the partition is not known in advance? This concern extends to systems with multiple partition coefficients. It would also be valuable to understand the seasonal variation of these coefficients, as I assume ET, for instance, varies seasonally. Furthermore, how was ET determined?

The mass-normalized breakthrough curve (mNBTC) refers to tracer mass, not water fluxes, as described throughout Section 3. It becomes analogous to water only for an ideal tracer. Even in that special case, however, it does not correspond to "the discharge at time $T + t_i$ generated by the input at time $t_i$," but rather to the *fraction of the input* at time $t_i$ that exits via discharge at time $t_i + T$, as explained on line 110 for the water breakthrough curve. To clarify this point, we will modify sentence on line 197 as follows:

"To derive the formula for the mass breakthrough curve in streamflow (i.e. the fraction of the input tracer mass found in streamflow at subsequent lag times), we proceed as we did with the water breakthrough curve […]"

The partitioning coefficient can be retrieved *a posteriori* by integrating the mNBTC over the entire age domain. In practice, this can be done reliably only after long-enough time has passed since the input time, by integrating over a sufficiently long age range until the integral converges to a plateau close to its asymptotic value (*sensu* Rigon et al. (2016) in the case of water partitioning). We have taken great care to ensure this condition is met. More details on the partitioning coefficient are provided in Section 5.4 and in Appendix B. We believe that the case of multiple partitioning coefficients is addressed in the same sections.

We agree that partitioning coefficients likely vary seasonally due to changes in hydrological fluxes and storage. As this partly explains the spread of values shown in Figure 5, we will add a mention to this effect on line 328 as follows:

"The spread of the partitioning coefficients in the box plots is primarily driven by variations in hydrological fluxes and storage, which may include a seasonal component, and is therefore largely influenced by the specific time series used in this study."

Evapotranspiration was retrieved from Duchemin et al. (2025) which computed fluxes based on a three-compartment model based on Kirchner (2016b, 2019) (see line 238), but we did not include additional methodological details for computing potential ET and ET because these fluxes are not intended to represent any specific real-world system. Potential ET is obtained from the Hargreaves method. ET is then obtained from the three-compartment model, where PET is multiplied by a linear soil-moisture stress function, with parameters derived from a Budyko equilibrium.

Page 9 - Line 233: The information that the models were implemented in Python is not essential. More relevant would be information about code availability and licensing. I consider code availability an important step toward understanding implementation details, which can be technically challenging.

We agree and will remove the mention of the programming language on line 233. Code availability is already noted at the end of this section on line 260. Upon revision, we will replace the current GitHub link with a permanent public repository (e.g., Zenodo) to ensure long-term accessibility, proper versioning, and clear licensing.

Page 9 - Line 239: The authors state they use the Kirchner 2016b model as a foundation. I would therefore expect the overall TTD model to address convolutions of two reservoirs, not just one. Have the authors done this? This point remains unclear and requires clarification. A treatment of this specific subject can be found, for instance, in Rigon and Bancheri (2021). That said, there is nothing inherently wrong with addressing a simpler system.

The Kirchner (2016b) model was used solely to generate realistic input and output fluxes; it was not used to compute transit times. These fluxes were then used to drive a single randomly sampled box model, deliberately avoiding convolutions of two or more reservoirs.

We emphasize that the focus of this study is on randomly sampled systems (which by definition are represented as single-box model), as stated for example on line 5 of the abstract, in Section 2.2 and in Figure 1. We will add a recall in Section 5 in the description of the numerical implementation, immediately after the mention of the Kirchner (2016b, 2019) model on line 239:
"While generated from a multiple-box model, these fluxes were subsequently used to drive a single randomly sampled box model in our study."

Also, we thank the Referee for pointing out the Rigon and Bancheri (2021). We will include it in the "Water age equations" section in the "Starting points" section.

References:
Duchemin, Q., Zanoni, M. G., Floriancic, M. G., Seybold, H., Obozinski, G., Kirchner, J. W., & Benettin, P. (2025). Data-driven estimation of the hydrologic response using generalized additive models. Geoscientific Model Development, 18(22), 8663-8678.

Kirchner, J. W. (2016b). Aggregation in environmental systems–Part 2: Catchment mean transit times and young water fractions under hydrologic nonstationarity. Hydrology and Earth System Sciences, 20(1), 299-328.

Kirchner, J. W. (2019). Quantifying new water fractions and transit time distributions using ensemble hydrograph separation: theory and benchmark tests. Hydrology and Earth System Sciences, 23(1), 303-349.

Rigon, R., Bancheri, M., & Green, T. R. (2016). Age-ranked hydrological budgets and a travel time description of catchment hydrology. *Hydrology and Earth System Sciences*, *20*(12), 4929-4947.

Rigon, R., & Bancheri. M. (2021). On the Relations between the Hydrological Dynamical Systems of Water Budget, Travel Time, Response Time and Tracer Concentrations. Hydrological Processes 35 (1).

---

## Author Comment (AC2)

**Response to Referee #2**

Please find below our responses to Ype van der Velde, with the original comments in black and our responses in blue.

This technical note describes how the transit time approach developed for tracking water ages through catchments can be extended to yield transit times of reactive tracers. I find this paper a well written and valuable addition to the theory development around transit time distributions of water and solutes. This study applies the theory of transit time distributions for water to solutes and combines it with assumptions from advective-dffusion solute transport modelling (Retardation: constant equilibrium between dissolved and sorbed concentration, linear decay, fractionation). As such it is a novel contribution.

We thank Ype van der Velde for his thorough review of our manuscript and appreciate the positive evaluation of our work.

Below I list a couple of suggestions that in my opinion could improve the manuscript:
Overall:
The paper has an unconventional structure: introduction, starting point, New solutions, insights from these new solutions, insights from Numerical implementation of these new solutions, conclusions. Consider a more conventional setup with: Introduction (1), Theoretical development (2, containing the old 2, 3 & 4), Example application (3) methods/setup (3.1), results (3.2), discussion (3.3) and conclusion (6). I can see why you chose your setup: you don't want to put a strong focus on the model implementation and it is not the goal in itself, but now section 5 reads cramped and its goal is unclear.

We agree that the structure is unconventional and may thus complicate the interpretation of the manuscript. We will restructure it as follows:

1. Introduction
2. Theoretical developments
    2.1 Starting points
        2.1.1   Water age equations (previously section 2.1)
        2.1.2   Tracer mass balance (previously section 2.2)
    2.2 New tracer transit time solutions (previously section 3)
3. Numerical implementation (previously introduction of section 5)
4. Results and Discussion
    4.1 Insights from the analytical expressions (previously section 4)
    4.2 Passive tracers (previously section 5.1)
    4.3 Non-passive tracers (previously section 5.2)
    4.4 Tracer breakthrough curves (previously section 5.3)
    4.5 Mass partitioning to multiple outputs (previously section 5.4)
5. Conclusions

Because this study presents a theoretical development supported by conceptual numerical experiments, we believe that the results and their interpretation are inherently interdependent. For this reason, we prefer to combine the Results and Discussion into a single section to maintain a continuous narrative flow, as is already done implicitly in the reviewed version of the manuscript.

In the theory development and discussion I miss what your results mean for not fully mixed (i.e. preferential flow systems) systems. It looks to me that without the full mixing-assumptions you can almost fully copy the same reasoning and your derivations will also hold for preferential flow systems. Could you not do the theory development for any system

and then do your example (section 5) for the fully mixed one? That would make the structure a bit easier to read as well? You can easily write the SoluteMass-Age balance eq:

$$\frac{\partial M(t-s,t)}{\partial t} = M_I(t)\delta(t-s=0) - \frac{M(t-s,t)}{RS(t-s,t)}Q(t-s,t) - \frac{\alpha M(t-s,t)}{RS(t-s,t)}E(t-s,t)$$
$$- k\,M(t-s,t)$$

Which solves to:

$$p_{Mq}(t-s,t) = \frac{M(t-s,t)}{M(t)} = \frac{M_I(s)}{M(t)}\exp\left(-\int_s^t \left[\frac{Q(t-s,\tau)+\alpha E(t-s,\tau)}{RS(t-s,\tau)}+k\right]d\tau\right)$$

We acknowledge that developments and discussion about the case of not randomly sampled systems are mostly absent. We deliberately avoided to discuss this case in detail, as our primary goal is to understand the key principles governing the discrepancies between water and tracer transit times. Extending the theoretical development to more general, non–randomly sampled systems would be an interesting new development, but we are concerned that pursuing a fully general formulation would introduce substantial technical complexity. This is primarily due to additional assumptions that would be required: how is mass applied during dry periods dissolved into waters of different ages? Is mass of a given age allowed to be transported by water of a different age? And if yes, how? These problems vanish in randomly sampled systems, where the mass age distribution remains the same regardless of how it is partitioned into waters of different ages.

The simpler case of mass applied only through a water input and always traveling along with that same water input can in principle be addressed using a general mass-age balance equation. However, we are not entirely sure that the solution proposed by the reviewer addresses the general case, as it seems to implicitly rely on a random sampling assumption and indeed it results in the classic exponential solution of randomly sampled systems.

That said, we agree that the underlying reasoning does not fundamentally rely on full mixing, and that the conceptual framework could, in principle, be extended to preferential flow systems with appropriate modifications. To make this explicit, we will add a clear statement to this effect in Section 3 of the first version of the manuscript.

In fact, we can only derive water transit time distributions through following tracers (=solutes) through the system. Thus basically, all previous publications that first derive Pq and subsequently use $C_Q(t) = \int_0^\infty C_J(t-T)p_Q(T,t)dT$ must have already implicitly applied a Solute transit time distributions, just with R=1, alfa =1 and k = 0 (stable isotopes). I think you can probably show both approaches are exactly the same thing. But you are much deeper into this and can reason better why or why not. I can be convinced that this all can be tackled through extra discussion on mixed systems.

Indeed, this convolution integral could be expressed in terms of backwards tracer TTD. We will add a paragraph on this point in the discussion when addressing the discrepancies in backwards TTDs of water and solutes.

I would suggest to write the conclusions more engaging less bullet-wise.

Thank you for the suggestion. We agree and will revise the Conclusions section to adopt a paragraph-based structure rather than a bullet-point format.

Line 223
"The evapoconcentration parameter α can be seen as a valve that controls the mass directed to ET. When α < 1, less mass will go to ET, resulting in tracer accumulation in storage and thus longer transit times. Conversely, when α > 1 (net tracer extraction) the larger tracer output rate will result in a depleted tracer storage, with faster turnover and shorter transit times." Although entirely correct, I find the valve quite abstract as plants don't have valve but can fractionate at their roots. I would suggest α < 1 for solutes that plants try to keep out of their system (high chloride, toxic solutes), α > 1 for solutes that plants need to for their functioning (Nutrients, trace elements)

We agree that the "valve" analogy is not a physically accurate representation of plant processes and could be misleading. We will modify this sentence as follows:

"The evapoconcentration parameter α can be seen as a regulator directing mass to ET. When α < 1, less mass will go to ET, resulting in tracer accumulation in storage and thus longer transit times. Conversely, when α > 1 (net tracer extraction) the larger tracer output rate will result in a depleted tracer storage, with faster turnover and shorter transit times."

We recall that a more physically grounded description of $\alpha$, explicitly framed in terms of selective uptake and exclusion processes, is already provided when the parameter is introduced (line 151). We therefore hope that sufficient context is now given for readers to understand that this modified analogy is not meant as a plant-physiological explanation of root-level processes, but rather as a simplified conceptual aid to interpret the model behavior.

Figure 5: Caption it would help add the partition coefficient variables to the caption (I was scanning the paper for θmR, but couldn't find it until close reading the caption)

Thank you for the suggestion. We will add the partition coefficient variables to the caption.

Conclusion 2 is not clear and to me not really a conclusion: "Such analytical solution builds on the framework initiated by Botter et al. (2010) and extends it to the case of reactive tracers"

We agree that this sentence is not sufficiently self-contained to stand as a separate conclusion. We will therefore integrate it into the introductory paragraph of the revised Conclusions section, where it will serve to summarize the theoretical foundations on which our work builds and the way our framework extends previous studies.

**Unrelated but I had too much fun adding a dissolution term. Maybe for next paper. Just for fun adding a source term that could represent weathering. Not sure if it works:**

$$\frac{\partial M(t-s,t)}{\partial t} = M_I(t)\delta(t-s=0) - \frac{M(t-s,t)}{RS(t-s,t)}Q(t-s,t) - \frac{\alpha M(t-s,t)}{RS(t-s,t)}E(t-s,t)$$
$$- k\,M(t-s,t) + j\left(\frac{M(t-s,t)}{RS(t-s,t)}\text{-C0}\right)$$

$$M(t-s,t) = M_I(s)\exp\left(-\int_{\tau=s}^{t}\left[\frac{Q(t-s,\tau)+\alpha E(t-s,\tau)-j}{RS(t-s,\tau)}+k\right]d\tau\right)$$
$$-jC_0\int_{\xi=s}^{t}\exp\left(-\int_{\tau=\xi}^{t}\left[\frac{Q(t-s,\tau)+\alpha E(t-s,\tau)-j}{RS(t-s,\tau)}+k\right]d\tau\right)d\xi$$

Thank you for these additional ideas! We gave it some thoughts and believe that freshly dissolved mass should, arguably, enter the system with age T=0. Thus, the input flux of dissolved mass $\dot{m}_d(t)$ would enter the tracer mass-age balance through a Dirac delta function, similarly to the input mass flux $\dot{m}_I(t)$. For a fully mixed system, the tracer mass age balance would then become:

$$\frac{d[M_s(t)\rho_s(T,t)]}{dt} = \dot{m}_I(t)\delta(T) + \dot{m}_d(t)\delta(T) - \dot{m}_Q(t)\rho_s(T,t) - \dot{m}_{ET}(t)\rho_s(T,t) + \dot{m}_R(t)\rho_s(T,t)$$

with $\dot{m}_d(t) = \lambda \cdot \left[\frac{M_s(t)}{R \cdot S(t)} - C_{eq}\right] \cdot S(t)$, where $\lambda$ is the kinetic constant and $C_{eq}$ corresponds to the immobile-phase equilibrium concentration (Maher, 2011).

This equation can be solved and yields:

$$\rho_s(T,t) = \left[\frac{\dot{m}_I(t-T)}{M_s(t)} - \frac{\lambda \cdot C_{eq} \cdot S(t-T)}{M_s(t)} + \frac{\lambda}{R}\right] \cdot exp\left[-\int_{t-T}^{t}\left(\frac{Q(\tau) + \alpha ET(\tau)}{R \cdot S(\tau)} + k\right)d\tau\right]$$

While this extension is mathematically feasible and conceptually interesting, it is limited to the case of dissolution only and does not account for solute precipitation when the equilibrium concentration is exceeded. Moreover, it introduces additional parameters, which would substantially complicate its interpretation. Therefore, we prefer to exclude it from the final manuscript.

Reference:
Maher, K. (2011). The role of fluid residence time and topographic scales in determining chemical fluxes from landscapes. *Earth and Planetary Science Letters, 312*(1-2), 48-58.